# Human Milk Feeding for Septic Newborn Infants Might Minimize Their Exposure to Ventilation Therapy

**DOI:** 10.3390/children9101450

**Published:** 2022-09-22

**Authors:** Elisenda Moliner-Calderón, Sergio Verd, Alfonso Leiva, Jaume Ponce-Taylor, Gemma Ginovart, Pia Moll-McCarthy, Catian Gelabert, Josep Figueras-Aloy

**Affiliations:** 1Neonatal Unit, Department of Paediatrics, Santa Creu i Sant Pau Hospital, 90 Mas Casanovas Street, 08041 Barcelona, Spain; 2Pediatric Unit, La Vileta Surgery, Department of Primary Care, Matamusinos Street, 07013 Palma de Mallorca, Spain; 3Balearic Islands Health Research Institute (IdISBa), 79 Valldemossa Road, 07120 Palma de Mallorca, Spain; 4Research Unit, Department of Primary Care, Escola Graduada Street, 07002 Palma de Mallorca, Spain; 5A & E Unit, Department of Primary Care, Illes Balears Street, 07014 Palma de Mallorca, Spain; 6Neonatal Unit, Department of Paediatrics, Germans Trias i Pujol Hospital, Canyet Road, 08916 Badalona, Spain; 7A & E Division, Manacor Hospital, Alcudia Road, 07500 Manacor, Spain; 8Department of Paediatrics, Son Espases Hospital, 79 Valldemossa Road, 07120 Palma de Mallorca, Spain; 9Neonatal Unit, ICGON, Clinic Hospital, Sabino Arana Street, 08028 Barcelona, Spain

**Keywords:** human milk, oxidative stress, oxygen therapy, premature infant, sepsis, newborn infant, ventilation support

## Abstract

Background. It has been well established that human milk feeding contributes to limiting lung diseases in vulnerable neonates. The primary aim of this study was to compare the need for mechanical ventilation between human milk-fed neonates with sepsis and formula-fed neonates with sepsis. Methods. All late preterm and full-term infants from a single center with sepsis findings from 2002 to 2017 were identified. Data on infant feeding during hospital admission were also recorded. Multivariate logistic regression analyses were performed to assess the impact of feeding type on ventilation support and main neonatal morbidities. Results. The total number of participants was 322 (human milk group = 260; exclusive formula group = 62). In the bivariate analysis, 72% of human milk-fed neonates did not require oxygen therapy or respiratory support versus 55% of their formula-fed counterparts (*p* < 0.0001). Accordingly, invasive mechanical ventilation was required in 9.2% of any human milk-fed infants versus 32% of their exclusively formula-fed counterparts (*p* = 0.0085). These results held true in multivariate analysis; indeed, any human milk-fed neonates were more likely to require less respiratory support (OR = 0.44; 95% CI:0.22, 0.89) than those who were exclusively formula-fed. Conclusion. Human milk feeding may minimize exposure to mechanical ventilation.

## 1. Introduction

Although it is known that the balance between oxidant status and antioxidant agents is associated with greater severity of preterm lung injury [1,2], there is a long way to go to achieve optimal management of the available preventive and therapeutic strategies for this condition [3]. Human milk (HM) consists of a spectacular array of biologically active components that facilitate optimal development and boost immunity in newborn infants. Many factors of HM, including macronutrients, oligosaccharides, and antioxidant and antimicrobial factors, play a fundamental immune-modulating role, and consequently contribute to protecting the immature lung [3]. Studies are currently underway to evaluate the roles of most of these molecular and cellular elements. According to these studies, antioxidants in HM appear to play a pivotal role in preventing lung injury [4].

Around birth, a delicate balance exists between the production of reactive oxygen species (ROS) and their detoxification by various biological systems [5]. At moderate concentrations, ROS play several beneficial roles in achieving healthy fetal or neonatal development. However, the balance may be disturbed due to the increased generation of ROS or inadequate detoxification, both of which are oxidative malfunctions inherent to many conditions affecting newborn infants, notably preterm birth, hyperoxia, and inflammation [6].

HM contains a considerable amount of antioxidant molecules [7]; antioxidants from HM can remove ROS directly or regulate signaling pathways to achieve their antioxidant actions [8]. Hence, newborn infants are heavily dependent on HM to counterbalance the oxidative stress that characterizes the different stages of the perinatal period. During labor, the neonate is subjected to a hyperoxic challenge [9]. There is a five-fold increase in oxygen tension from the intrauterine hypoxic environment to the extrauterine normoxic environment. After birth, neonates with sepsis or preterm infants have reduced antioxidant defenses. Pathogenic invasion represents the initiation of sepsis, but sepsis syndrome is subsequently maintained by a cascade of oxidative mechanisms and a marked disruption of the antioxidant metabolic pathways [10], which, once activated, act independently from the pathogens themselves. However, because premature birth interrupts antioxidant defense mechanisms that mature during late gestation, preterm infants are also highly sensitive to oxidative stress [11].

Oxidative insult is a salient part of lung injury that begins as acute inflammation in respiratory distress disease and can evolve into structural scarring. Hyperoxia directly causes alveolar epithelial apoptosis, which leads to aberrant airway remodeling [8]. While there are conflicting findings in the field of neonatal respiratory support, it is generally accepted that previous lung inflammation makes the lung endothelium more susceptible to oxidant-induced injury during mechanical ventilation [12].

Given that HM possesses a much stronger antioxidant potential than bovine infant formulas [13] and the reported antioxidant depletion among preterm or septic neonates, this study aimed to analyze the impact of HM feeding on lung protection or type of mechanical ventilation in neonates with sepsis. Since a recent Cochrane provides only low-certainty evidence about the effects of feeding preterm infants hydrolysate [14], and a very recent consensus document accepts that there is no data to support soy formula use in preterm infants [15], our study is limited to analyzing the role of bovine infant formulas.

## 2. Methods

### 2.1. Study Population, Design, Location, and Period

This analysis was conducted using data from a retrospective study on neonates diagnosed with sepsis. A detailed description of the study design and methods has been published previously [16]. This is a secondary analysis of a review of the electronic medical records of newborn infants from the neonatal intensive care unit (NICU) of Sant Pau Hospital, which is a tertiary referral unit in the province of Barcelona, Spain. The unit consists of 10 intensive care units and 7 high-care beds, with approximately 350 admissions per year and a bed occupancy rate of 90%.

A cohort study was undertaken in which all late preterm infants (all infants born at a gestational age between 34 weeks and 0 days, and 36 weeks and 6 days) and full-term infants up to 28 days of age with clinical and laboratory findings of bacterial or viral sepsis between January 2002 and December 2017 were identified. The beginning of this period corresponds to the time when the electronic medical record was implemented in our center, and the end of this period corresponds to the time when this study was planned. The computerized system of the neonatal unit provided retrospective data, including maternal parity and gravida, maternal diseases such as gestational diabetes, hypertension or chorioamnionitis, infants’ demographic and perinatal characteristics, feeding type, and 1 min and 5 min Apgar scores.

Data regarding infant feeding during the hospital stay were recorded as human milk, formula, or combined feeding. For the purpose of the study, we determined the following definitions for neonatal feeding: “any human milk” feedings when any mother’s own milk ± donor milk was administered or “exclusively formula” when feeding included all meals of this type.

### 2.2. Primary Outcome

Our aim was to determine whether exclusive formula feeding could predict respiratory support practices. Infant feeding type (any human milk vs. exclusive formula) was an independent variable and type of respiratory support (no support, Oxygen therapy, non-invasive ventilation, invasive mechanical ventilation) was the dependent variable in the initial analysis.

### 2.3. Secondary Outcomes

The secondary outcomes were neonatal characteristics, morbidities, and procedures according to the feeding type. These included fetal status at birth, acidosis, hypotension, abnormal neurological examination, abnormal brain scan, meningitis, positive blood cultures, CPAP and/or oxygen administration, ventilation type, discharge weight, and days of hospital stay.

### 2.4. Analysis Plan

Descriptive statistics were used to produce counts and percentages regarding the type of feeding of newborn infants with a history of neonatal sepsis in the registry. The primary and secondary outcomes and infant and maternal characteristics were compared using Fisher’s exact test and χ2 or *t*-tests, depending on the variable type. *p* values were obtained from bivariate comparisons as a function of each individual risk factor.

Multivariate logistic regression analyses were performed to assess the impact of feeding type on the primary outcome and to determine the independent contribution of each factor to neonatal outcome. Based on the statistical significance in bivariate comparisons, factors that were highly predictive of the primary outcome were included in the model to adjust for potential confounding risks. These covariates were removed from the multivariate logistic regression model via backward selection if they no longer showed significance when added to the model. Heart failure and respiratory failure were not included in the adjusted model because of their collinearity with respiratory support. No methods were used to adjust for potential bias.

SPSS software (version 21.0; SPSS Inc., Chicago, IL, USA) was used for the statistical analysis and data management. 

### 2.5. Ethical Considerations

This study was approved by the Clinical Studies Ethics Committee of Santa Creu i Sant Pau University Hospital on 20 December 2020 (decision number: IIBSP-ENT-2020-152). Since the study was conducted retrospectively, spanning over a period of two decades, and the data were extracted from patient files, no informed consent was obtained.

## 3. Results

A total of 25,152 infants were born alive in our delivery unit between 2002 and 2017, and 4210 (16.7%) were admitted to our NICU during the study period. Finally, 322 (7.6%) admitted neonates met the inclusion for participation in the final analysis. In Table 1, we report the patients’ baseline demographics for the two feeding-type groups: neonates who received any amount of HM and those who received only formula (EF). We found no differences in maternal-perinatal morbidity, gestational age, sex, 5 min Apgar score, length, or head circumference at birth. EF infants were more likely to have been delivered by C-section and had lower birth weight and 1 min Apgar scores than any HM-fed infants. However, these relationships did not persist when introduced into the multivariate models.

Table 2 compares the short-term neonatal outcomes according to feeding type. Bivariate analysis showed that EF infants required higher respiratory support and higher fraction of inspired oxygen (FiO_2_) levels than any HM-fed infants (0.28 versus 0.39, *p* = 0.0001). In addition, 72% of any HM-fed neonates did not require oxygen therapy or respiratory support compared to 55% of their EF-fed counterparts (*p* < 0.0001). Accordingly, invasive mechanical ventilation was required in 9.2% of any HM-fed infants versus 32% of their EF-fed counterparts (*p* = 0.0085). Other short-term neonatal outcomes did not differ between the two groups.

In the adjusted multivariate logistic regression model (Table 3), any HM-fed newborn infants were more likely to require less respiratory support (adjusted OR = 0.449; 95% CI:0.225, 0.893) than EF-fed newborn infants. Gestational age, altered alertness state, altered muscle tone, sepsis score, meningitis, and abnormal brain scan were covariates that remained significant in the final model. Important confounders in the adjusted model were gestational age and meningitis, which were associated with feeding type (*p* = 0.056 and *p* = 0.084, respectively) and type of respiratory support. However, even when controlling for these covariates, 55% (adjusted OR, 0.449) of newborns had lower odds of receiving invasive respiratory support than EF-fed newborns.

## 4. Discussion

This study demonstrated that EF-fed septic late preterm or full-term newborn infants require higher respiratory support than their any HM-fed counterparts. More than 100 studies have analyzed the preventive role of HM feeding on bronchopulmonary dysplasia [17,18]. However, to the best of our knowledge, no more than five papers in the last 20 years have addressed the link between HM feeding and lower ventilation requirements of critically ill neonates [19,20,21,22,23].

In 2005, Schanler et al. [19] reported that the mean duration of mechanical ventilation in 92 extremely preterm formula-fed neonates was 19 days, in contrast to 12 days for their 70 counterparts fed on their mothers’ own milk (*p* = 0.03). Ten years later, a study by Marinelli et al. [20] related to the implementation of a donor milk policy in a NICU reported that the number of days on supplemental oxygen therapy decreased to the same degree as the proportion of diet as HM increased within a group of 150 very-low-birth-weight infants. Over the last seven years, three studies have confirmed that increasing the amount of HM improves short-term respiratory support outcomes. In 2015, our own multicenter pre–post retrospective study, before and after implementing a donor human milk policy, found that the time in oxygen and duration of mechanical ventilation were significantly higher among EF-fed infants than among HM-fed infants [21]. Hair et al. [22] found a significant reduction in ventilator days among newborn infants with a birth weight < 1250 g on a new feeding protocol of mother’s own milk fortified with HM-derived fortifier compared to their counterparts who received a diet of mother’s own milk fortified with bovine fortifier and/or preterm formula. Very recently, Sun et al. [23] conducted a prospective study including infants born at <30 weeks’ gestation, in which mothers of 98 neonates in the intervention group were asked to provide at least one feed per day of fresh HM, and the control group included 109 mothers who did not agree to provide fresh HM but agreed that their infants received donor milk or frozen mother’s own milk. They found that the fresh HM group had a shorter duration of mechanical ventilation than did the control group. Finally, a 2022 meta-analysis showed that not only HM feeding but also oral care of preterm infants with mother’s milk shortens the mechanical ventilation time [24]. It is remarkable that despite large differences in the way various research groups provide HM to infants in neonatal care, HM is associated with reduced respiratory support in all cases. Moreover, our results on the need for mechanical ventilation are in line with previous research, despite focusing on late preterm or full-term neonates, unlike the cases described above. Consequently, it remains unclear what the optimal dose, type, and duration of HM is and which populations of neonates with sepsis or preterm infants might benefit most from HM supplementation to minimize exposure to mechanical ventilation.

HM is a bioactive factory with stem cells, microbiota, growth factors, numerous antioxidants, and immune-boosting properties [25], and its protective role against acute lung disease in infants is well established [26,27]; however, the mechanism of this beneficial effect is unclear. Several potential explanations can be advanced to clarify why we report an increased susceptibility to severe acute lung disease in EF-fed infants compared to those who were HM-fed. In particular, antioxidants are among the most abundant compounds found in HM [28,29,30], especially when the target population comprises a group of infected neonates who register a significant elevation of ROS. In this case, the risk of oxidative injury may persist and even increase throughout a septic neonate’s hospital course, making the antioxidant components of HM beyond the first days of life potentially beneficial. Furthermore, the imbalance in the redox status favoring oxidative pathway activation during sepsis [31] is also a critical factor in the pathophysiology of neonatal respiratory distress [32]. This hypothesis assumes that because HM is a better scavenger of ROS than formula [33], it may defend a large population of neonates with increased oxidative stress, a critical factor that exacerbates perinatal morbidities. The list of antioxidants in breast milk that potentially protect against neonatal chronic lung disease includes more than ten elements (probiotics, phytochemicals, amino acids, trace elements, vitamins, Glutathione, oligosaccharides, catalase, melatonin, lactoferrin, among others) [8]. However, many other HM biofactors are involved and thus contribute to neonatal lung protection. Notably, observations of oral care with mother’s milk in recent years suggest that HM can form a protective layer of the respiratory wall and play a first-line defense role by reducing oropharyngeal pathogens in ventilated neonates. On the other hand, HM non-digestible oligosaccharides facilitate the development of the early microbiome [34], which may have a beneficial effect on respiratory immunity [35]. However, up to now the influence of HM on the preterm airway tract microbiome is still an open question. Alternatively, HM protection against acute respiratory disease may be conferred by passive transfer of any soluble molecule with anti-infective properties [36]. Finally, there is still much to be learned about the immune-modulating role of the microbiota or the expanding field of HM stem cells.

## 5. Limitations

The large number of included septic infants and the exclusion of recall bias concerning HM feeding are the strengths of our study. We were able to show some variation in mechanical ventilation duration between any HM-fed and EF-fed newborn infants. While these findings are important for broadening our knowledge, they are not without limitations.

The sample was taken from the NICU of one hospital over a long period of time; thus, further studies are needed to determine whether these results generalize to other settings. The feeding type was not the focus of the primary study. Consequently, the questions concerning HM feeding were less detailed than those used in a prospective study. An additional limitation of this study was the use of maternal HM feeding at any time during hospital admission to represent the provision of HM. We were also unable to quantify the volume of HM received or distinguish between fresh mother’s milk and processed human milk. Our retrospective review design has inherent limitations, including missing data and a lack of standardization of measurement methods. Although the independent association between any HM feeding and less need for mechanical ventilation in septic neonates is supported by our findings, conclusions about causality cannot be established owing to the observational design of the study.

## 6. Conclusions

This study shows that EF feeding is independently associated with higher respiratory support not only among very preterm neonates but also among late preterm or full-term neonates with sepsis. These results reinforce the theory of HM as a postnatal intervention able to mitigate metabolic and immune-related risk factors of respiratory pathologies and add plenty of evidence on the ability of HM to counterbalance oxidative stress in newborn infants. As this is a purely observational cohort focusing on neonatal morbidity, it is not possible at this stage to identify specific changes in management that could potentially have led to improved outcomes over time. Rather, our results are hypothesis generating and must be confirmed with prospectively designed studies.

## Figures and Tables

**Table 1 children-09-01450-t001:** Background information.

Sample Characteristics	Study Groups		*p* Value ^a^
	Any Human Milk Feeding, N = 260	Exclusive Formula Feeding, N = 62	
Gestational hypertension	21 (80.7)	6 (9.7)	0.607
Gestational diabetes	8 (3.0)	2 (3.3)	1.000
Chrioamnionitis	63 (24.0)	20 (32.2)	0.218
Group B strep positive mother	36 (13.8)	5 (8.3)	0.512
Gestational age, weeks	38.59 (2.08)	38.01 (2.27)	0.056
Delivery type: vaginal/C-section	72 (27.4)	(46.6)	0.0052 **
Multiple gestation	13 (4.9)	7 (13.2)	0.071
Apgar 1 min	8 (2)	7 (3)	0.036 *
Apgar 5 min	9 (1)	9 (2)	1.000
Birth weight, g	3116 (639)	2893 (637)	0.010 *
Height, cm	48.6 (2.84)	48.22 (2.69)	0.322
Cranial circumference, cm	33.83 (2.02)	33.37 (1.94)	0.093
Girls/boys	126/136	26/34	0.567

Note: Data expressed as number (%) or mean (standard deviation). ^a^ Comparison between neonates with any human milk feeding and exclusive formula feeding. * *p* < 0.05; ** *p* < 0.01.

**Table 2 children-09-01450-t002:** Clinical outcomes by feeding type.

	Any Human Milk Feeding N = 260	Exclusive Formula Feeding, N = 62	*p* Value ^a^
Apnea	47 (18.0)	17 (27.4)	0.074
Abnormal alertness state	143 (55.0)	27 (43.5)	0.198
Abnormal muscle tone	118 (45.3)	28 (45.1)	0.886
Convulsions	10 (3.8)	4 (6.4)	0.304
Hypotension	43 (16.5)	16 (25.8)	0.094
Neonatal acidosis	77 (29.6)	24 (40.6)	0.121
Positive blood culture, n (%)	81 (31.1)	18 (29.0)	1.000
Meningitis	20 (10.1)	0/31 (0)	0.084
Type of respiratory support			
- No support	190 (72)	34 (55)	
- Oxygen therapy	14 (5.3)	3 (4.8)	
- Noninvasive ventilation	34 (13)	4 (6.5)	
- Invasive mechanical ventilation	24 (9.2)	19 (31)	0.000067 ***
Fraction of inspired Oxygen	0.28 (0.16)	0.39 (0.25)	0.0001 **
Hospitalization, days	10 (9)	12 (9)	0.121
Abnormal brain ultrasound	13 (4.6)	1 (1.9)	0.482
Discharge weight, g	3425 (689)	3256 (639)	0.083

Note: Data expressed as number (%) or mean (standard deviation). ^a^ Comparison between neonates with any human milk feeding and exclusive formula feeding. ** *p* < 0.01; *** *p* < 0.001.

**Table 3 children-09-01450-t003:** Multivariate logistic regression analysis of factors associated with higher respiratory support.

Variable	Odds Ratio	Standard Error	z	*p* > |z|	95% CI
Human milk feeding	0.449	0.157	−2.28	0.023	0.225 to 0.893
Weeks of gestation	0.821	0.051	−3.12	0.002	0.725 to 0.929
Sepsis score	3.668	1.430	3.33	0.001	1.708 to 7.877
Abnormal alertness state	1.576	0.334	2.15	0.032	1.040 to 2.388
Abnormal muscle tone	3.239	1.089	3.49	0.000	1.675 to 6.261
Meningitis	0.505	0.126	−2.73	0.006	0.310 to 0.825
Abnormal brain ultrasound	5.473	2.491	3.73	0.000	2.242 to 13.358

Abbreviations: CI, Confidence Interval; *p* > |z|, Corresponding Probability for the Reduced Logistic Model; z, Estimated Z-Score.

## Data Availability

The datasets generated during and/or analyzed during the current study are available from the corresponding author upon reasonable request.

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
