# Peer review of "Human Milk Feeding for Septic Newborn Infants Might Minimize Their Exposure to Ventilation Therapy"

_children, 2022, doi:10.3390/children9101450_

Round 1
Reviewer 1 Report
The article is well done I believe that the introduction can be more exhaustive as regards the contextualization of the topic, certainly references must be added in particular for the first paragraph. Some references cited in the discussion (2003/2001/2009) are also not very recent.
Author Response
REVIEWER 1
Thank you so much for taking the time to post a positive review of our study. Please see below, for a response to your comments.
The article is well done I believe that the introduction can be more exhaustive as regards the contextualization of the topic, certainly references must be added in particular for the first paragraph. Some references cited in the discussion (2003/2001/2009) are also not very recent.
ANSWER: We agree. First paragraph of the Introduction has been expanded to provide a more adequate background for this paper and to include references up to date. In addition, we have added references from the last 8 years to the Discussion.
Reviewer 2 Report
Manuscript Number: Children-1900446
Title: Human milk feeding for septic newborn infants might minimize their exposure to ventilation therapy
This study examined the impact of any human milk feeding vs. exclusive infant formula feeding on the need for respiratory therapy in neonates diagnosed with sepsis. This manuscript addresses an important area, understanding the impact of human milk on neonate health outcomes, and is of interest to the readership of Children. However, a several areas should be addressed before publication.
Thank you for the opportunity to review this paper.
Specific Comments/Questions:
Abstract
Authors may consider defining “sick neonates” (e.g., it would clearer to use “neonates with sepsis”).
Authors may consider clarifying “… impact of the type of milk….” as infant formula is not considered milk (e.g., it would be clearer to list human milk or infant formula) here and throughout. The Methods indicate that the human milk category was any human milk (regardless of quantity); Authors may consider using “any human milk” here and throughout.
Introduction
The Introduction is well-written and could be strengthened by including additional references throughout. For example, in the first paragraph, a sentence starts with “According to these studies, …” but there are no references for readers to refer to.
It would strengthen the Introduction to discuss the antioxidant properties (or lack thereof) for infant formula. It might also be important to note that although the majority of infant formula is made from bovine-milk proteins, there are infant formulas that do not include these proteins.
Authors may consider clarifying in the research aim that this study looks at neonates with sepsis.
Method
Line 83: It would be helpful to define late preterm for readers.
Line 85: Was the computerized system the electronic health record for each patient?
Line 88-89: Authors may consider defining “maternal diseases” and “early clinical features”.
Line 97-98: This is different from the research aim stated in the introduction. Could remove this sentence.
Line 98-99: Authors may consider defining feeding type here (e.g., Infant feeding type (any human milk vs. exclusive formula) …”) for clarity. Authors may also consider defining the dependent variable, respiratory support, here (e.g., was this any type of respiratory support?).
General: Authors mention the data was abstracted between 2002-2017. Authors may consider discussing why this range was selected (e.g., sample size driven?). Might medical advances in the treatment of neonatal sepsis decrease the need for respiratory support across this time range? Did the Authors conduct any sensitivity analyses to see if time impact the results?
General: Did the Authors conduct any tests using a three category independent variable (i.e., exclusive human milk, human milk and infant formula, and exclusive infant formula)? This might be informative as feeding human milk once or twice would likely confer less benefit over feeding exclusive human milk. This might not have been possible due to small sample sizes.
Results
Table 1: There are a few typographical errors that the Authors should fix prior to publication. It might also be informative to report late preterm and full terms results here.
Discussion/Conclusion
Line 176-177: Can the Authors provider references for the five papers mentioned here.
Author Response
REVIEWER 2
We are very grateful to Reviewer 2 for this provision of feedback on our manuscript.
Please see below, for a point-by-point response to your comments.
This study examined the impact of any human milk feeding vs. exclusive infant formula feeding on the need for respiratory therapy in neonates diagnosed with sepsis. This manuscript addresses an important area, understanding the impact of human milk on neonate health outcomes, and is of interest to the readership of Children. However, a several areas should be addressed before publication.
Thank you for the opportunity to review this paper.
Specific Comments/Questions:
Abstract
Authors may consider defining “sick neonates” (e.g., it would clearer to use “neonates with sepsis”).
ANSWER. We have turned sick neonates into neonates with sepsis throughout the text.
Authors may consider clarifying “… impact of the type of milk….” as infant formula is not considered milk (e.g., it would be clearer to list human milk or infant formula) here and throughout.
ANSWER. We have erased the word milk linked to formula throughout the text.
The Methods indicate that the human milk category was any human milk (regardless of quantity); Authors may consider using “any human milk” here and throughout.
ANSWER. Now we refer always to any human milk regarding this group of patients.
Introduction
The Introduction is well-written and could be strengthened by including additional references throughout. For example, in the first paragraph, a sentence starts with “According to these studies, …” but there are no references for readers to refer to.
ANSWER. Updated references have been added to this sentence and to the rest of the Introduction.
It would strengthen the Introduction to discuss the antioxidant properties (or lack thereof) for infant formula. It might also be important to note that although the majority of infant formula is made from bovine-milk proteins, there are infant formulas that do not include these proteins.
ANSWER. We refer to: Lugonja, N.; Spasić, S.D.; Laugier, O. et al. Differences in direct pharmacologic effects and antioxidative properties of mature breast milk and infant formulas. Nutrition 2013, 29, 431-435. And we have added a paragraph on the restrictions on the use of hydrolysates or soy-derived formula in preterm infants to justify that this study only considered the bovine-derived formula.
Authors may consider clarifying in the research aim that this study looks at neonates with sepsis.
ANSWER: Done.
Method
Line 83: It would be helpful to define late preterm for readers.
ANSWER. Late pretem definition added to the text.
Line 85: Was the computerized system the electronic health record for each patient?
ANSWER. Yes
Line 88-89: Authors may consider defining “maternal diseases” and “early clinical features”.
ANSWER. Maternal diseases have been defined. Early clinical features have been turned into 1-minutes and 5-minutes Apgar scores.
Line 97-98: This is different from the research aim stated in the introduction. Could remove this sentence.
ANSWER. We have removed this sentence.
Line 98-99: Authors may consider defining feeding type here (e.g., Infant feeding type (any human milk vs. exclusive formula) …”) for clarity.
ANSWER. Accordingly, we have defined feeding type
Authors may also consider defining the dependent variable, respiratory support, here (e.g., was this any type of respiratory support?).
ANSWER. We have defined type of respiratory support: no support, Oxygen therapy, non-invasive ventilation, invasive mechanical ventilation.
General: Authors mention the data was abstracted between 2002-2017. Authors may consider discussing why this range was selected (e.g., sample size driven?).
ANSWER. We have added this sentence: The beginning of this period corresponds to the time when the electronic medical record was implemented in our centre, and the end of this period corresponds to the time when this study was planned.
Might medical advances in the treatment of neonatal sepsis decrease the need for respiratory support across this time range?
ANSWER. In the Limitations section we take into account that the long period of time conditions our results, although we do not compare the type of respiratory support from one moment to another, but according to the type of feeding received.
Did the Authors conduct any sensitivity analyses to see if time impact the results?
ANSWER. No, we did not.
General: Did the Authors conduct any tests using a three category independent variable (i.e., exclusive human milk, human milk and infant formula, and exclusive infant formula)? This might be informative as feeding human milk once or twice would likely confer less benefit over feeding exclusive human milk. This might not have been possible due to small sample sizes.
ANSWER. Our data do not allow us to make these comparisons. We were unable to distinguish between exclusive human milk, human milk and infant formula, and exclusive infant formula.
Results
Table 1: There are a few typographical errors that the Authors should fix prior to publication. It might also be informative to report late preterm and full terms results here.
ANSWER. Table 1, typographical errors corrected.
We cannot distinguish between late preterm and full terms results here.
Discussion/Conclusion
Line 176-177: Can the Authors provider references for the five papers mentioned here.
ANSWER. Done.
Reviewer 3 Report
Authors have done a good study that may have clinical relevance in the NICU.
In this study, authors have concluded that human milk is beneficial in decreasing the need of mechanical ventilation in sick infants with sepsis in comparison to formula fed infants. Although it is a small study, it is good to see these results. I have a few minor comments:
1. Table 1 has background information and parenthesis has % or mean. Specifying which one of percentage and mean would be helpful. For gestational hypertension, 80.7 seems % but should be 8.07%
Author Response
REVIEWER 3
We are very grateful to Reviewer for reviewing our manuscript.
Authors have done a good study that may have clinical relevance in the NICU.
In this study, authors have concluded that human milk is beneficial in decreasing the need of mechanical ventilation in sick infants with sepsis in comparison to formula fed infants. Although it is a small study, it is good to see these results. I have a few minor comments:
1. Table 1 has background information and parenthesis has % or mean. Specifying which one of percentage and mean would be helpful. For gestational hypertension, 80.7 seems % but should be 8.07%
ANSWER. Table 1 corrected. We apologyze for this mistake.
Reviewer 4 Report
A very well written and comprehensive paper. Few suggestions would be to
1. The authors mention "Detailed description of the study design and methods is 77 published [10]." in line 77. It would help the reader to refer to a table, may be in the supplementary data.
2. Suggest mentioning the authors in
"Ten years later, a paper related 180 to the implementation of a donor milk policy in a NICU reported that the number of days 181 on supplemental oxygen therapy decreased to the same degree as the proportion of diet 182 as HM increased within a group of 150 very-low-birth-weight infants [13]." (line 180)
3. Its mentioned that antioxidants in HM might be acting as ROS scavengers and providing the protective effect. It would be good to have a table listing the known antioxidants in milk.
[check ref 4 in the manuscript (https://doi.org/10.3389/fnut.2022.924036 )]
Author Response
REVIEWER 4
We are grateful for Reviewer's 4 comments which have undoubtedly contributed to improving our study.
A very well written and comprehensive paper. Few suggestions would be to
1. The authors mention "Detailed description of the study design and methods is 77 published [10]." in line 77. It would help the reader to refer to a table, may be in the supplementary data.
ANSWER. We refer now to Tables 2&6 of this reference.
2. Suggest mentioning the authors in
"Ten years later, a paper related 180 to the implementation of a donor milk policy in a NICU reported that the number of days 181 on supplemental oxygen therapy decreased to the same degree as the proportion of diet 182 as HM increased within a group of 150 very-low-birth-weight infants [13]." (line 180)
ANSWER. We have added the name of the authors to the text
3. Its mentioned that antioxidants in HM might be acting as ROS scavengers and providing the protective effect. It would be good to have a table listing the known antioxidants in milk.
[check ref 4 in the manuscript (https://doi.org/10.3389/fnut.2022.924036 )]
ANSWER. We have added this list to the last paragraph of the Discussion.